# Methylene Blue-Loaded Mesoporous Silica-Coated Gold Nanorods on Graphene Oxide for Synergistic Photothermal and Photodynamic Therapy

**DOI:** 10.3390/pharmaceutics14102242

**Published:** 2022-10-20

**Authors:** Sun-Hwa Seo, Ara Joe, Hyo-Won Han, Panchanathan Manivasagan, Eue-Soon Jang

**Affiliations:** Department of Applied Chemistry, Kumoh National Institute of Technology, Gumi 730-701, Gyeongbuk, Korea

**Keywords:** methylene blue, gold nanorods, graphene oxide, diagnosis, phototherapy

## Abstract

Photo-nanotheranostics integrates near-infrared (NIR) light-triggered diagnostics and therapeutics, which are combined into a novel all-in-one phototheranostic nanomaterial that holds great promise for the early detection and precise treatment of cancer. In this study, we developed methylene blue-loaded mesoporous silica-coated gold nanorods on graphene oxide (MB-GNR@mSiO_2_-GO) as an all-in-one photo-nanotheranostic agent for intracellular surface-enhanced Raman scattering (SERS) imaging-guided photothermal therapy (PTT)/photodynamic therapy (PDT) for cancer. Amine functionalization of the MB-GNR@mSiO_2_ surfaces was performed using 3-aminopropyltriethoxysilane (APTES), which was well anchored on the carboxyl groups of graphene oxide (GO) nanosheets uniformly, and showed a remarkably higher photothermal conversion efficiency (48.93%), resulting in outstanding PTT/PDT for cancer. The in vitro photothermal/photodynamic effect of MB-GNR@mSiO_2_-GO with laser irradiation showed significantly reduced cell viability (6.32%), indicating that MB-GNR@mSiO_2_-GO with laser irradiation induced significantly more cell deaths. Under laser irradiation, MB-GNR@mSiO_2_-GO showed a strong SERS effect, which permits accurate cancer cell detection by SERS imaging. Subsequently, the same Raman laser can focus on highly detected MDA-MB-23l cells for a prolonged time to perform PTT/PDT. Therefore, MB-GNR@mSiO_2_-GO has great potential for precise SERS imaging-guided synergistic PTT/PDT for cancer.

## 1. Introduction

Cancer is a leading cause of human death globally, and exploiting photo-nanotheranostics is an urgent requirement for cancer treatment [1,2]. Photo-nanotheranostics is the integration of near-infrared (NIR) light-triggered diagnostics and therapeutics, which are combined into a new all-in-one nanomaterial that has great potential for accurate cancer diagnosis and effective phototherapy [3]. As an emerging photo-triggered diagnostic technique, surface-enhanced Raman scattering (SERS) imaging has been increasingly considered a promising diagnostic method for the development of theranostic systems because of its high sensitivity, enhanced photostability, and multiplexing abilities [4]. SERS has emerged as an effective imaging method for accurate cancer detection [5]. As emerging phototherapeutic strategies, photothermal therapy (PTT) and photodynamic therapy (PDT) have attracted increasing clinical attention in recent years, which is beneficial for enhanced cancer therapy because of their synergistic effects [6]. PTT has attracted considerable interest in cancer treatment because of its minimal invasiveness, high efficiency, low toxicity, and deep tissue penetration [7,8]. In PTT, a photothermal nanoagent directly converts the absorbed NIR light into heat-to-kill cancer cells [9]. PDT has attracted much attention as a less invasive therapeutic modality for treating cancer. It employs photosensitizers (PSs) and light irradiation at certain wavelengths of photon energy to produce reactive oxygen species (ROS), including singlet oxygen (^1^O_2_), to destroy cancer cells [7]. Therefore, the integration of SERS imaging, PTT, and PDT treatment into one system could provide a new photo-nanotheranostic platform for precise cancer diagnosis and therapy [6].

To date, various nanoparticles (NPs) with strong NIR absorption properties and an effective heat transfer mechanism have been developed as photo-nanotheranostic agents for SERS imaging-guided phototherapy, including polymer NPs [10], carbon NPs [11], copper sulfide NPs [12], and gold nanostructures [13]. Among these photothermal nanomaterials, gold nanorods (GNRs) have emerged as interesting nanomaterials for theranostic applications, including drug delivery, PTT, SERS imaging, and photoacoustic imaging, because of their excellent small size, biocompatibility, ease of synthesis, ease of surface modification, and strong tunable extinction in the NIR region (600–1100 nm) [14]. The seed-mediated method has become one of the most powerful and versatile methods for the synthesis of GNRs due to its advantages, such as a simple process and high yield of GNRs [15,16]. GNRs have proven to be appropriate candidates for SERS imaging-guided PTT because of their strong SERS effects and high photothermal conversion properties [17]. However, GNRs with nonporous structures have the poor dye-loading capacity, which seriously limits their efficacy in PDT [18]. The Stöber method is a widely used method for obtaining silica-coated NPs due to its advantages, such as ease of synthesis, low cost, mild reaction conditions, and a broad range of achievable particle sizes [19,20]. Mesoporous silica (mSiO_2_) is highly suitable as a coating material for GNRs [21]. Mesoporous silica-coated gold nanorods (GNR@mSiO_2_) show great potential for theranostic applications because of their high specific surface area, high accessibility, tunable pore size and volume, high dye loading capability, high stability, well-defined surface properties, and excellent biocompatibility [22]. Methylene blue (MB) is a water-soluble, Food and Drug Administration (FDA)-approved, and photosensitizer (PS) drug that is frequently used in PDT to treat cancerous and non-cancerous diseases [23]. MB has high levels of ROS, such as singlet oxygen (^1^O_2_) quantum yield, under laser light exposure [24]. MB was directly loaded with high loading capability inside the pores of mSiO_2_ to obtain MB-GNR@mSiO_2_ [25]. Most importantly, the integration of GNRs with other NPs, such as graphene oxide (GO) is being actively pursued because of its enhanced photothermal conversion property when compared with single NPs [3]. GO is a sheet-like two-dimensional layer of carbon nanomaterials and is the most attractive material for the PTT of cancer because of its excellent biocompatibility, conductivity, large surface area, stability, facile surface functionalization, and high optical absorption coefficient in the NIR region [26]. Therefore, the integration of MB-GNR@mSiO_2_ with GO is a promising strategy for achieving high PTT efficacy because of its synergistic effect [9]. In this study, we developed multifunctional MB-GNR@mSiO_2_-GO as a novel photo-nanotheranostic agent for SERS imaging-guided synergistic PTT/PDT.

## 2. Materials and Methods

### 2.1. Materials

All the materials and reagents were purchased from Sigma-Aldrich (St. Louis, MO, USA) and used without further purification.

### 2.2. Synthesis of MB-GNR@mSiO_2_-GO

Gold nanorods (GNRs) were prepared in an aqueous cetyltrimethylammonium bromide (CTAB) solution using a seed-mediated method [15]. For the mesoporous silica (mSiO_2_) coating on the surface of GNRs (GNR@mSiO_2_), GNR@mSiO_2_ was synthesized according to a previously published procedure, with minor modifications [27,28]. Briefly, 10 mL of the purified GNR solution was added to 5 mL of absolute ethanol and 5 mL of deionized water (DW), and the mixture was stirred for 3 h at ambient temperature. Subsequently, 200 µL of NaOH solution (0.1 M) was slowly added dropwise with stirring. Subsequently, 60 µL of tetraethyl orthosilicate (TEOS; 20%) in methanol solution was added three times at 30 min intervals with gentle stirring. The mixture was allowed to react for 2 days at 30 °C with vigorous stirring. Mesoporous silica-coated GNRs (GNR@mSiO_2_) were collected by centrifugation at 15,000 rpm for 30 min and washed thrice with ethanol. The purified GNR@mSiO_2_ samples obtained were redispersed in 10 mL of ethanol for further experiments. Methylene blue (MB) loaded onto GNR@mSiO_2_ was prepared according to previous reports [25]. MB (0.1 mg) was added to 10 mL of purified GNR@mSiO_2_, and the reaction solution was then stirred for 2 days at 25 °C, which was covered with aluminum foil to avoid light exposure and to protect the fluorescent dyes from photodegradation. The reaction solutions (MB-GNR@mSiO_2_) were separated by centrifugation and washed repeatedly with DW until the supernatant changed from dark blue to colorless.

For the conjugation of amine functional groups on the surface of MB-GNR@mSiO_2_, 1 mL of 3-aminopropyltriethoxysilane (APTES) was mixed with 10 mL of MB-GNR@mSiO_2_ and stirred for 6 h at ambient temperature. The reaction solutions were centrifuged and washed thrice using DW. The aminated MB-GNR@mSiO_2_-APTES was re-dispersed in DW for further experiments. GO was prepared using a modified Hummers’ method [29]. The primary amino groups of MB-GNR@mSiO_2_-APTES (5.0 mL) were reacted with the carboxyl groups of GO (0.1 mg) via a coupling reaction in the presence of 1-ethyl-3-(3-dimethylaminopropyl) carbodiimide hydrochloride (EDC; 1.0 mg) and *N*-hydroxysuccinimide (NHS; 1.0 mg) with overnight stirring at room temperature. The amino group of MB-GNR@mSiO_2_-APTES can be easily cross-linked with the carboxyl group of GO, and MB-GNR@mSiO_2_-GO can potentially prevent the leaching of MB from the mesopores of SiO_2_. The reaction solutions (MB-GNR@mSiO_2_-GO) were centrifuged, washed repeatedly using DW to eliminate unreacted reactants, and freeze-dried.

### 2.3. Characterization

The UV–vis spectra of GNR, GNR@mSiO_2_, MB-GNR@mSiO_2_, free GO, and MB-GNR@mSiO_2_-GO were obtained using a Shimadzu (Kyoto, Japan) UV-2600 UV–vis spectrophotometer. The morphologies of the nanomaterials were characterized using a Cs-corrected field emission transmission electron microscope (Cs-corrected FE-TEM; JEM-ARM-200F, JEOL, Tokyo, Japan) equipped with energy-dispersive X-ray spectroscopy (EDX). The particle size range and zeta potential (ZP) of the nanomaterials were measured using a particle size and ZP analyzer (90 Plus, Brookhaven Instruments, Holtsville, NY, USA). Raman spectra at 785 nm (2 mW, spot area = 2–100 µm) were obtained using a Renishaw inVia Raman spectrometer (Gloucestershire, UK) equipped with a Leica DM 2500 microscope with 1200 g/mm grating. The final gold (Au) ion concentration of the nanomaterials in aqueous solutions was determined to be 36 µg Au/mL by inductively coupled plasma optical emission spectroscopy (ICP-OES, Santa Clara, CA, USA). The photothermal experiments were carried out using an NIR laser at 785 nm (Changchun New Industries Optoelectronics Technology, Changchun, China; power density: 0.8 W/cm^2^, laser beam diameter: 4 mm, laser beam shape: round), and infrared (IR) thermal images were captured every 1 s using an FLIR CX-Series compact thermal imaging camera. A transparent cuvette was filled with a known concentration of NP solutions and the top surface was fixed with a laser pointer lens. An IR thermal camera was placed at a 30 cm distance to capture a lateral image of the cuvette and a laser.

### 2.4. Measurement of Photothermal Performance

One milliliter of each of the GNR, GNR@mSiO_2_, MB-GNR@mSiO_2_, GO, and MB-GNR@mSiO_2_-GO aqueous solutions (36 µg/mL) was added to a quartz cuvette and exposed to a 785 nm NIR laser at 0.8 W/cm^2^ for 25 min. The temperature profiles were recorded every 1 s using an FLIR CX-Series compact thermal imaging camera. Additionally, the gold concentration of the MB-GNR@mSiO_2_-GO was calculated to be 36 µg Au/mL and different volumes (0–200 µL of 36 µg/mL) of the MB-GNR@mSiO_2_-GO aqueous solutions (1 mL) were added to a quartz cuvette and exposed to a 785 nm laser at 0.8 W/cm^2^ for 25 min. Temperature profiles and infrared (IR) thermal images were obtained using a FLIR thermal imaging camera. To evaluate the photothermal stability of MB-GNR@mSiO_2_-GO, the MB-GNR@mSiO_2_-GO (200 µL of 36 µg/mL) aqueous solution was filled in a quartz cuvette and then exposed to four cycles of laser on/off (785 nm, 0.8 W/cm^2^). The UV–vis spectra of MB-GNR@mSiO_2_-GO were recorded and characterized using TEM, DLS, and ZP after four cycles of laser irradiation. The photothermal conversion efficiency (*η*) of MB-GNR@mSiO_2_-GO (200 µL of 36 µg/mL) was calculated according to the thermal balance using the following Equation (1), as described in previous reports [30,31].
(1)$$\eta =\frac{hS\left(T_{Max}-T_{Sur}\right)-Q_{dis}}{I\left(1-10^{-A_{808}}\right)}$$

### 2.5. Cells Culture

The mouse muscle fibroblast cell line (BLO-11) and human breast cancer cell line (MDA-MB-231 cells) were purchased from the Korean Cell Line Bank (Seoul, Korea). The BLO-11 and MDA-MB-231 cells were cultured in Dulbecco’s modified Eagle’s medium (DMEM) with 1% antibiotics (penicillin/streptomycin) and 10% fetal bovine serum (FBS) at 37 °C under 5% CO_2_.

### 2.6. In Vitro PTT/PDT Synergistic Effect

The in vitro photothermal/photodynamic effects of GNR, MB-GNR@mSiO_2_, GO, and MB-GNR@mSiO_2_-GO against normal (BLO-11) and cancer (MDA-MB-231) cells were evaluated using the 3-(4,5-dimethylthiazol-2-yl)-2,5-diphenyltetrazolium bromide (MTT) assay. BLO-11 and MDA-MB-231 cells (7000 cells/well) were seeded in 96-well plates and cultured at 37 °C for 24 h. The cells were incubated with GNR, MB-GNR@mSiO_2_, GO, and MB-GNR@mSiO_2_-GO (200 µL of 36 µg/mL) for 12 h and 48 h and then irradiated with a 785 nm laser at 0.8 W/cm^2^ for various time points (0, 10, 30, and 50 min). The cells were treated with all the nanomaterials and were incubated with a 785 nm laser irradiation as the control. All cells were further incubated for 6 h, and cell viability was evaluated using the MTT assay. The percentage of cell viability was determined by the following Formula (2):(2)$$\mathrm{Percentage}\;\mathrm{of}\;\mathrm{cell}\;\mathrm{viability}\;\left(\%\right)=\frac{\mathrm{OD}\;\mathrm{value}\;\mathrm{of}\;\mathrm{samples}}{\mathrm{OD}\;\mathrm{value}\;\mathrm{of}\;\mathrm{controls}}\times 100\;$$

To further confirm the in vitro photothermal/photodynamic effect of GNR, MB-GNR@mSiO_2_, and MB-GNR@mSiO_2_-GO, the MDA-MB-231 cells were stained using calcein-AM and propidium iodide (PI) to visually identify live (green) and dead (red) cells. The MDA-MB-231 cells were cultured in 6-well plates and incubated overnight at 37 °C. After incubation, the cells were treated with GNR, MB-GNR@mSiO_2_, and MB-GNR@mSiO_2_-GO (200 µL of 36 µg/mL) for 12 h and irradiated with a 785 nm laser (0.8 W/cm^2^) for various time points (0, 10, 30, and 50 min). Then, all cells were incubated for 6 h and stained using calcein-AM and PI for 30 min. Finally, the cells were imaged using a confocal fluorescence microscope (Nikon Confocal A1R; Nikon, Tokyo, Japan).

To quantify intracellular ROS, MDA-MB-231 cells were cultured in 6-well plates and incubated for 24 h at 37 °C. The MDA-MB-231 cells were then incubated with GNR, MB-GNR@mSiO_2_, and MB-GNR@mSiO_2_-GO (200 µL of 36 µg/mL) for 12 h and exposed to a 785 nm laser at 0.8 W/cm^2^ for various time points (0, 10, 30, and 50 min). The cells were incubated for another 6 h and stained with 2,7-Dichlorofuorescin diacetate (DCFH-DA 10 µM) for 30 min. DCFH-DA is a non-fluorescent dye used to detect PDT-induced intracellular ROS production. The cells were washed with PBS and imaged using a confocal fluorescence microscope (Nikon Confocal A1R; Nikon, Tokyo, Japan).

### 2.7. SERS Imaging Measurement

The SERS imaging measurement was recorded with a Renishaw inVia Raman spectrometer (Gloucestershire, UK) equipped with a NIR laser (785 nm, 2 mW, spot area = 2–100 µm) and a Leica DM 2500 microscope with 1200 g/mm grating, which was focused for 40 s on the samples through a 20× microscope objective. MDA-MB-231 cells were grown on a chambered slide glass at 37 °C for 24 h and then cells were treated with GNR (control), MB-GNR@mSiO_2_, and MB-GNR@mSiO_2_-GO. After incubation, the slide glass was performed using a Renishaw inVia Raman microscope.

### 2.8. Statistical Analysis

Data were expressed as mean ± standard deviation (SD) in at least three trials. The significant difference between groups was performed using a Student’s two-tailed *t*-test and one-way ANOVA with SPSS software 23.0. * *p* < 0.05 and ** *p* < 0.01 were statistically significant.

## 3. Results and Discussion

### 3.1. Synthesis and Characterization of MB-GNR@mSiO_2_-GO

The preparation process of MB-GNR@mSiO_2_-GO for PTT/PDT is illustrated in Figure 1. GNRs were prepared in an aqueous CTAB solution using a seed-mediated growth method [15]. The synthesized GNRs were covered with a mesoporous layer of silica (mSiO_2_) using a previously reported method, forming highly monodisperse GNR@mSiO_2_ [27,28]. MB was directly loaded inside the pores of mSiO_2_ with a high loading capacity to obtain MB-GNR@mSiO_2_ [25]. Amine functionalization of the MB-GNR@mSiO_2_ surfaces was performed using APTES as a silane coupling agent (MB-GNR@mSiO_2_-APTES). GO was synthesized using a modified Hummers’ method [29]. Finally, the carboxyl groups of GO were covalently conjugated to the free amino groups of MB-GNR@mSiO_2_-APTES through a coupling reaction in the presence of EDC/NHS chemistry to form MB-GNR@mSiO_2_-GO, which can potentially prevent leaching of the loaded MB molecules before MB-GNR@mSiO_2_-GO is taken up by the cells [32].

The morphologies of GNR, GNR@mSiO_2_, MB-GNR@mSiO_2_, GO, and MB-GNR@mSiO_2_-GO were characterized using Cs-corrected FE-TEM (Figure 1). The GNRs have a uniform and well-defined rod-like structure with an average length of 47 ± 3.6 nm and a diameter of 12 nm ± 2.4 nm. The GNRs were coated with mSiO_2_ to form highly monodispersed GNR@mSiO_2_. The thickness of the mSiO_2_ was 38 nm, and a nanoporous structure could be observed. Similar findings were reported by Xia et al., who demonstrated that the thickness of the GNR@mSiO_2_ was 12 nm [33]. After loading with MB, MB-GNR@mSiO_2_ exhibited a uniform size and morphology. The as-prepared GO was nanosheet-shaped with a smooth surface and wrinkled edges. MB-GNR@mSiO_2_ was uniformly anchored on the GO nanosheets (Appendix A). Additionally, Cs-corrected FE-TEM and the corresponding EDX mapping of MB-GNR@mSiO_2_-GO revealed the presence of Au, Si, S, C, and O, confirming the presence of MB-GNR@mSiO_2_-GO (Figure 2).

Figure 3a shows the UV–vis spectra of GNR, GNR@mSiO_2_, MB-GNR@mSiO_2_, GO, and MB-GNR@mSiO_2_-GO. The as-prepared GNR has a strong longitudinal surface plasmon resonance (LSPR) peak at 800 nm, which is more suitable for PTT with an NIR laser [34]. After coating with SiO_2_, the LSPR peak of GNR@mSiO_2_ was slightly red-shifted to 810 nm, which could be attributed to an increase in the local refractive index of the surrounding medium after the creation of the silica shell layer by replacing the CTAB [35]. After loading with MB, the absorption spectra of MB-GNR@mSiO_2_ showed a characteristic peak of MB at 655 nm and a strong LSPR peak at 800 nm, indicating that MB molecules were successfully loaded onto GNR@mSiO_2_ and were highly effective in PDT, owing to their ability to generate ROS [36]. The absorption spectrum of GO spanned from the UV-vis range to the NIR range, which can effectively enhance the photothermal conversion efficiency [37]. The absorption spectra of MB-GNR@SiO_2_-GO exhibited characteristic peaks of GO, MB, and GNR@SiO_2_ at 280, 655, and 810 nm, respectively, confirming their strong photothermal and photodynamic properties.

The particle sizes of GNR, GNR@mSiO_2_, MB-GNR@mSiO_2_, GO, and MB-GNR@mSiO_2_-GO were measured by dynamic light scattering (DLS). The GNR, GNR@mSiO_2_, MB-GNR@mSiO_2_, GO, and MB-GNR@mSiO_2_-GO have average sizes of 43.67 ± 2.18 nm, 188.46 ± 3.42 nm, 196.57 ± 4.17 nm, 80.47 ± 4.02 nm, and 294.90 ± 2.74, respectively (Figure 3b). The average particle size of GNR@mSiO_2_ increased to ~144.79 nm when compared with GNR, indicating that the mSiO_2_ layer was effectively coated on the surface of the GNR. The average particle size of MB-GNR@mSiO_2_ also increased to ~8.11 nm compared with GNR@mSiO_2_, demonstrating that MB was successfully loaded onto GNR@mSiO_2_. Additionally, the particle size of MB-GNR@mSiO_2_-GO increased significantly to ~98.33 nm, implying that GO was covalently conjugated to MB-GNR@mSiO_2_. As shown in Figure 3c, the zeta potentials (ZP) of GNR, GNR@mSiO_2_, MB, MB-GNR@mSiO_2_, GO, and MB-GNR@mSiO_2_-GO were measured in PBS (pH 7.4). The ZP of GNR was +35.4 ± 2.0 mV due to the presence of positively charged CTAB on the surface of GNR. The ZP of GNR@mSiO_2_ was −31.0 ± 1.5 mV because of the hydroxyl groups on the surface of GNR@mSiO_2_. The ZP of MB was +7.5 ± 0.4 mV because of the positively charged blue dye. After MB loading, the ZP of MB-GNR@mSiO_2_ was −31.4 ± 2.0 mV because the positively charged MB can be easily absorbed on the mSiO_2_, which indicated the successful loading of MB and GNR@mSiO_2_. The ZP of GO was −34.0 ± 2.0 mV because of the ionized carboxylic acid groups. The ZP of MB-GNR@mSiO_2_-GO was −39.0 ± 2.0 mV which confirmed the successful conjugation of GO and MB-GNR@mSiO_2_-GO [38].

The micro-Raman spectra of the synthesized GNR, MB-GNR@mSiO_2_, and MB-GNR@mSiO_2_-GO were analyzed using an excitation laser at 785 nm (2 mW, spot area = 2–100 µm) and are shown in Figure 3d. Strong Raman signals were recorded for MB-GNR@mSiO_2_ and MB-GNR@mSiO_2_-GO, while GNR did not show any Raman signals. The Raman spectra of MB-GNR@mSiO_2_ and MB-GNR@mSiO_2_-GO showed intense bands at 246 cm^−1^, 445 cm^−1^, 771 cm^−1^, 1317 cm^−1^, 1397 cm^−1^, and 1624 cm^−1^, assigned to the Au–N stretching mode (ν(Au–N)), C–N–C skeletal deformation mode (δ(C–N–C), γ(C–H) stretching mode, ν(N–CH_3_) mode, ν_sym_(C–N) mode, and ν(C–C)_ring_ mode, respectively. The Raman spectra of MB-GNR@mSiO_2_-GO were much stronger than those of MB-GNR@mSiO_2_, suggesting that GO has a high binding affinity for the MB-GNR@mSiO_2_ surface [25,28,39].

### 3.2. Measurement of Photothermal Performance

To further assess the photothermal efficacy of MB-GNR@mSiO_2_-GO, we examined the temperature variations of GNR, GNR@mSiO_2_, MB-GNR@mSiO_2_, GO, and MB-GNR@mSiO_2_-GO aqueous solutions (36 µg/mL) under a 785 nm laser irradiation at a power density of 0.8 W/cm^2^ for 25 min (Figure 4a). The temperature variation of GNRs, GNR@mSiO_2_, MB-GNR@mSiO_2_, and GO aqueous solutions increased from 36.42 °C to 41.28 °C, which was insufficient to ablate the cancer cells. Meanwhile, the temperature variation of pure water only increased from 36.24 °C to 27.28 °C. In contrast, the temperature variation of the MB-GNR@mSiO_2_-GO aqueous solutions increased to 50.67 °C, which is sufficient for cancer therapy because cancer cells can be specifically eliminated at temperatures higher than hyperthermia (42–47 °C) [40]. Additionally, the Au concentration of MB-GNR@mSiO_2_-GO was calculated to be 36 µg Au/mL, and different volumes (0–200 µL of 36 µg/mL) of MB-GNR@mSiO_2_-GO aqueous solutions (1 mL) were added to a quartz cuvette and exposed to a 785 nm laser at 0.8 W/cm^2^ for 25 min. The temperature variation of MB-GNR@mSiO_2_-GO (0, 50, 100, 150, and 200 µL of 36 µg/mL) aqueous solutions were exposed to a 785 nm laser at 0.8 W/cm^2^ for 25 min, resulting in temperature values of 37.4 °C, 45.79 °C, 48.72 °C, 49.71 °C, and 52.12 °C, respectively (Figure 4b). The photothermal effect of the MB-GNR@mSiO_2_-GO (0, 50, 100, 150, and 200 µL of 36 µg/mL) aqueous solutions was further examined using a FLIR thermal imaging camera (Figure 4c).

The photothermal conversion efficiency (*η*) is a significant parameter used to quantify the ability of photothermal agents to convert light into heat [41]. To further demonstrate the photothermal conversion efficiency (*η*) of the MB-GNR@mSiO_2_-GO aqueous solution, 1 mL of the MB-GNR@mSiO_2_-GO (200 µL of 36 µg/mL) aqueous solution was exposed to a 785 nm laser at 0.8 W/cm^2^ for 25 min, turned off for 25 min, and then left to cool down to room temperature (Figure 5a). The value of *t_s_* was 398.81 s. The photothermal conversion efficiency (*η*) of MB-GNR@mSiO_2_-GO (200 µL/mL) was calculated to be 48.93% (Figure 5b), confirming that MB-GNR@mSiO_2_-GO has the best photothermal conversion capability, which was higher than that of commonly reported photothermal materials, such as GNRs (21%) [42], GO nanosheets (18%) [9], Fe_3_O_4_@CuS NPs (19.2%) [43], Cu_2–x_S nanocrystals (16.3%) [44], and Fe_3_O_4_ nanoclusters (20.8%) [45]. More interestingly, the photothermal conversion efficiency of MB-GNR@mSiO_2_-GO was significantly higher (48.93%) than that of either GNRs (21%) alone or GO nanosheets (18%) alone, which strongly supports the idea that the dual plasmonic photothermal agents have a synergistic plasmonic effect [9,46].

It is widely known that GNR and GO have excellent heat-conducting properties, which are crucial for enhancing the photothermal stability of MB-GNR@mSiO_2_-GO (200 µL of 36 µg/mL) [47]. To further evaluate the photothermal stability of MB-GNR@mSiO_2_-GO, the solution was placed in a quartz cuvette and then exposed to four cycles of laser on/off at 0.8 W/cm^2^. As shown in Figure 5c, no significant changes in the temperature variation were observed for MB-GNR@mSiO_2_-GO after four laser on/off cycles, indicating that MB-GNR@mSiO_2_-GO has good photothermal stability. Additionally, the UV–vis spectra of MB-GNR@mSiO_2_-GO were recorded after four cycles of laser irradiation (Figure 5d). The UV–vis spectra demonstrated that MB-GNR@mSiO_2_-GO exhibited no obvious changes in absorbance after four cycles of laser irradiation. MB-GNR@mSiO_2_-GO was also characterized using TEM, DLS, and ZP after four cycles of laser irradiation. The FETEM image of MB-GNR@mSiO_2_-GO also showed a uniform size and morphology after four cycles of laser irradiation (Figure 5e). The particle size of MB-GNR@mSiO_2_-GO after four cycles of laser irradiation was characterized by DLS as ~292.75 ± 2.92 nm. The ZP of MB-GNR@mSiO_2_-GO was −36.81 ± 3.31 mV. These results indicate that MB-GNR@mSiO_2_-GO has excellent photothermal stability after four cycles of laser irradiation [47].

### 3.3. In Vitro PTT/PDT Synergistic Effect

To evaluate the in vitro photothermal/photodynamic synergistic efficacy of GNR, MB-GNR@mSiO_2_, GO, and MB-GNR@mSiO_2_-GO, the BLO-11, and MDA-MB-231 cells were incubated with GNR, MB-GNR@mSiO_2_, GO, and MB-GNR@mSiO_2_-GO (200 µL of 36 µg/mL) for 12 h and 48 h, and then were exposed to a 785 nm laser at 0.8 W/cm^2^ for various time points (0, 10, 30, and 50 min). All cells were further incubated for 6 h, and their viability was determined by the MTT assay. As shown in Appendix A, BLO-11 cells showed no obvious cytotoxicity without laser irradiation and the cell viability of BLO-11 cells with laser irradiation for 10, 30, and 50 min was above 80% when the high concentration of GNR, MB-GNR@mSiO_2_, and GO was 200 µL of 36 µg/mL for 12 h and 48 h, indicating that GNR, MB-GNR@mSiO_2_, and GO have excellent biocompatibility. The cell viability with laser irradiation for 10 and 30 min was above 80% at a higher concentration of MB-GNR@mSiO_2_-GO (200 µL of 36 µg/mL) and the cell viability with laser irradiation for 50 min was more than 75% at a high concentration of MB-GNR@mSiO_2_-GO up to 200 µL of 36 µg/mL, suggesting very little cytotoxicity and good biocompatibility for the MB-GNR@mSiO_2_-GO. As shown in Figure 5f and Appendix A, no noticeable cytotoxicity was observed in any of the treated groups for 12 h and 48 h of incubation without laser irradiation, suggesting that the as-prepared GNR, MB-GNR@mSiO_2_, GO, and MB-GNR@mSiO_2_-GO were highly biocompatible. However, upon irradiation, GNR, MB-GNR@mSiO_2_, and GO showed noticeable anticancer activity against MDA-MB-231 cells, which was 70.58%, 58.55%, and 73.57% for 12 h, and was 64.41%, 56.32%, and 71.42% for 48 h at the high concentration of GNR, MB-GNR@mSiO_2_, and GO up to 200 µL of 36 µg/mL and exposed to a 785 nm laser at 0.8 W/cm^2^ for 50 min. The cells were treated with MB-GNR@mSiO_2_-GO (200 µL of 36 µg/mL) for 12 h and 48 h and then were irradiated with a 785 nm laser at 0.8 W/cm^2^ for 50 min, which showed significantly reduced cell viability (6.32% and 4.46%), demonstrating the possibility that laser irradiation could boost the cytotoxic effects of MB-GNR@mSiO_2_-GO [34]. As shown in Figure 5f and Appendix A, no significant difference was observed after 12 h and 48 h incubation with GNR, MB-GNR@mSiO_2_, GO, and MB-GNR@mSiO_2_-GO (200 µL of 36 µg/mL) with or without laser irradiation, confirming that all the materials have very little toxicity to cells without laser irradiation; in addition, all the materials with laser irradiation for 50 min showed significant cytotoxicity to MDA-MB-231 cells because these materials have excellent photothermal conversion properties.

The photothermal effects of GNR, MB-GNR@mSiO_2_, and MB-GNR@mSiO_2_-GO on MDA-MB-231 cells were further confirmed using calcein-AM and PI co-staining to visually differentiate live (green) and dead (red) cells. As shown in Figure 6, there were no cellular deaths in the control only, control with laser irradiation, GNR only, GNR with laser irradiation, MB-GNR@mSiO_2_ only, MB-GNR@mSiO_2_ with laser irradiation, and MB-GNR@mSiO_2_-GO only groups because all of the cells showed green fluorescence and negligible cell deaths in these groups. In contrast, MB-GNR@mSiO_2_-GO with laser irradiation showed strong red fluorescence, and a large red area was observed, suggesting that MB-GNR@mSiO_2_-GO with laser irradiation induced complete cellular destruction [48].

To evaluate the photodynamic effect of GNR, MB-GNR@mSiO_2_, and MB-GNR@mSiO_2_-GO on intracellular ROS generation, DCHF-DA was used as a fluorescent dye to monitor changes in the intracellular ROS levels. As shown in Figure 7, no strong green signal was detected in the control only, control with laser irradiation, GNR only, GNR with laser irradiation, MB-GNR@mSiO_2_ only, MB-GNR@mSiO_2_ with laser irradiation, and MB-GNR@mSiO_2_-GO only groups, which did not damage the cancer cells. In contrast, MB-GNR@mSiO_2_-GO produced strong green fluorescence after 785 nm laser irradiation for 50 min, which could damage biomolecules and cellular organs such as the mitochondria and nucleus, implying the generation of ROS [49].

### 3.4. Surface-Enhanced Raman Scattering (SERS) Imaging

Surface-enhanced Raman scattering (SERS) imaging is a technique for boosting the Raman signals of molecules using nanomaterials, such as metal colloids [50]. SERS has drawn much interest in the field of biological research and has emerged as an effective imaging method for early cancer detection [51]. The bright-field images and SERS spectra of the GNR (control), MB-GNR@mSiO_2_, and MB-GNR@mSiO_2_-GO-treated MDA-MB-231 cells under an excitation laser of 785 nm are shown in Figure 8a–c. Strong SERS peaks were observed for MB-GNR@mSiO_2_ and MB-GNR@mSiO_2_-GO, whereas no SERS peaks were observed for the control. The SERS peaks of MB-GNR@mSiO_2_ and MB-GNR@mSiO_2_-GO were clearly observed at 445 cm^−1^, 1397 cm^−1^, and 1624 cm^−1^, which were assigned to the C–N–C skeletal deformation mode (δ(C–N–C), ν_sym_(C–N), and ν(C–C)_ring_ modes, indicating that the SERS peaks of MB-GNR@mSiO_2_-GO were sufficient to detect cancer at an early stage.

## 4. Conclusions

In summary, we successfully designed and developed multifunctional MB-GNR@mSiO_2_-GO as a new phototheranostic agent for intracellular SERS imaging-guided synergistic PTT/PDT of cancer. MB-GNR@mSiO_2_-GO showed strong SPR absorption in the NIR region, high MB loading capability, high photothermal conversion efficiency (48.93%), superior photothermal stability, and a strong SERS effect. The in vitro cytotoxicity of MB-GNR@mSiO_2_-GO had excellent biocompatibility in the absence of laser irradiation. Most intriguingly, the strong SERS effect of MB-GNR@mSiO_2_-GO permits accurate cancer cell detection by SERS imaging, and the cells treated with MB-GNR@mSiO_2_-GO and laser irradiation led to significant cell death, confirming that MB-GNR@mSiO_2_-GO has a strong SERS effect and superior photothermal efficiency under laser irradiation. The in vitro photodynamic effect of MB-GNR@mSiO_2_-GO showed enhanced cell apoptosis under laser irradiation, which could damage the mitochondria and nucleus, indicating that MB-GNR@mSiO_2_-GO has excellent ROS generation capability with the help of 785 nm laser irradiation. We believe that this multifunctional MB-GNR@mSiO_2_-GO has great potential for further clinical applications in the diagnosis and therapy of cancer.

## Data Availability

Not applicable.

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
