# Peer review of "Methylene Blue-Loaded Mesoporous Silica-Coated Gold Nanorods on Graphene Oxide for Synergistic Photothermal and Photodynamic Therapy"

_pharmaceutics, 2022, doi:10.3390/pharmaceutics14102242_

Round 1
Reviewer 1 Report
The manuscript is interesting and important dealing with combined supramolecular assemblies for diagnostic and therapy against cancer.
A few points need to be considered for revision:
1)Title shortening by eliminating "for surface-enhanced Raman scattering "
2)In the abstract: a) explain how MB-GNR@mSiO2 was anchored on the GO; b)if cell viability was reduced to 7.92% there was not complete cell destruction, please, correct this in the abstract; c)explain in the abstract why the combination induced a strong SERS effect also in the abstract
3)Line 35: check spelling; define NIR
4)Define all undefined abbreviations in the Introduction
5)Give principles of methods used in the Introduction
6)In the Introduction briefly explain how gold nanorods were coated by silica
7)Line 68: replace “poor”by “small”. Why small would be important
8)Define all undefined abbreviations in the Figure Caption of each figure
9)Figure 1, line 227, GO cannot be visualized in the ;ast subfigure
10)Add readable dimension bars
11)Lines 262-269: approximations are required for zeta-potentials. For example 35.43 +/- 1.77 is 35 +/-2
12)Figure 3(c): is it possible to determine the zeta-potential of MB? Does MB aggregate in water solution?
13)The reader would benefit of a better explanation on principles of each method employed (mainly SERS). Add this to M&M
14)Lines 453-455: rephrase for clarity and also correct grammar: …more ROS… What is meant by this? More than what?
Author Response
The authors thanks to reviewer #1 for his/her valuable and positive comments. The corrections were made in the revised manuscript as per the reviewer’s suggestion

Reviewer 2 Report
In the manuscript "Methylene blue-loaded mesoporous silica-coated gold nanorods on graphene oxide for surface-enhanced Raman scattering imaging-guided synergistic photothermal therapy/photodynamic therapy" the authors are reporting in vitro results regarding the synergistic photothermal therapy/photodynamic therapy for methylene blue-loaded mesoporous silica-coated gold nanorods against MDA-MB-231 cancer cells.
The results reported present potential for treatment of cancer cells. However, there are some issues that need to be improved.
In material and methods section the author have to better describe the SERS method. Also, the authors need to say the filter cubes used to take the images as well as the objective used.
The author are reporting the MTT assay for different experimental conditions against MDA-MB-231 cancer cells, however the authors are not discussing the effects of the same conditions against normal cells or haemolysis.
The authors mention a synergistic effect. How did they get to this conclusion? There are several methods to calculate this. Which one was used. This have to be presented in the material and methods section.
The authors need to report also the number of replica for each experiments and statistical difference for the data.
Author Response
The authors thanks to the Editor’s for his/her valuable and positive comments. The corrections were made in the revised manuscript as per the reviewer's suggestion

Reviewer 3 Report
In this manuscript, the authors have prepared a multifunctional NPs (Methylene blue-loaded mesoporous silica-coated gold nanorods on graphene oxide) for imaging-guided synergistic photothermal therapy/photodynamic therapy. The authors have described how to prepare and characterize the formulate NPs, and in my opinion, this study (pharmaceutics-1907615) provides fundamental info in this field, and could be accepted after clarify the long-term toxicity and considering the following comments:
1. The authors should provide the longer cell viability measurement (48 h or 72 h) for the control samples in order to evaluate the long-term toxicity of their prepared NPs (the NPs and laser only might be toxic for a long term incubation, and induce side-effects).
2. The toxicity of the control group with the graphene oxide only should be provided with and without laser.
3. The laser beam size and shape of the beam need to be mentioned. Not clear that the laser beam was focused on the NPs or was at the shoulders of the Gaussian beam (far away from the focus area).
4. In Fig. 1, the scale bars are not clear.
5. The unit of 200 µL/mL does not make sense, and the applied/used concentration (200 µL of 36 µg/mL?) is required, and it needs to be corrected.
Author Response
The authors thanks to reviewer #1 for his/her valuable and positive comments. The corrections were made in the revised manuscript as per the reviewer’s suggestion.

Round 2
Reviewer 2 Report
The authors have made the changes necessary and the quality of the paper improved. The paper can be published in the present form
Reviewer 3 Report
accept